# OpenReview forum: "Large Language Models Are Not Strong Abstract Reasoners"
_ICLR.cc/2024/Conference — Submitted to ICLR 2024_

### Official Review · Reviewer_As6m · 2023-11-01

**Soundness:** 2 fair
**Presentation:** 3 good
**Contribution:** 2 fair
**Rating:** 5
**Confidence:** 4

**Summary:**

This paper presents an evaluation of Large Language Models (LLMs) on abstract reasoning tasks. The authors introduce a new benchmark for evaluating LLMs on abstract reasoning and conduct extensive experiments on language models. The results show that LLMs currently achieve limited performance on abstract reasoning tasks compared to other natural language tasks. The authors also explore the impact of fine-tuning and prompt design techniques on abstract reasoning performance.

**Strengths:**

1. This article attempts to address a topic of great interest - whether large models possess the capacity for abstract reasoning.
2. The authors provide a comprehensive evaluation and conduct extensive experiments on various language models.

**Weaknesses:**

1. Similar conclusion has been explored by previous studies [1][2].

[1] "Reasoning or reciting? exploring the capabilities and limitations of language models through counterfactual tasks." arXiv preprint arXiv:2307.02477 (2023).

[2] "Large Language Models are In-Context Semantic Reasoners rather than Symbolic Reasoners." arXiv preprint arXiv:2305.14825 (2023)

2. Lack of experiment with larger models or advanced models. Fine-tuned on smaller models cannot sufficiently draw the conclusion.

**Questions:**

1. Can you experiment with more advanced models Llama-2, with better performance than Llama1, Alpaca, or fine-tune with larger models (13B, 70B)?
2. The details of fine-tuning experiments, such as training data, training steps. Do you consider incorporating the instruction about “how to induce”, “how to deduce” into supervision?

---

> ### Author Response · Authors · 2023-11-16
>
> Thank you for your constructive feedback!
>
> > Similar conclusion has been explored by previous studies [1][2].
> > [1] "Reasoning or reciting? exploring the capabilities and limitations of language models through counterfactual tasks." arXiv preprint arXiv:2307.02477 (2023).
> > [2] "Large Language Models are In-Context Semantic Reasoners rather than Symbolic Reasoners." arXiv preprint arXiv:2305.14825 (2023)
>
> Yes, some papers have recently attempted to tackle similar problems than ours. Our study has been independently conducted and released at the same time as [1,2].
>
> Regarding [1], the authors measure counterfactual tasks, defined as tasks close from one in the training data but with one uncommon setting (e.g. arithmetic in base 9). While this is related to our work, this is different from a measure of the ability of LLMs to extract and reason on abstract patterns. We also provide an additional analysis of how using different prompting strategies, text/symbolic formats, open-ended or multiple-choices affect the models' performance, and what kind of tasks are the most affected.
>
> Regarding [2], the authors measure the logical reasoning abilities of LLMs via inductive, deductive and abductive reasoning tasks. While this is related to our work, the paper measures an ability different from abstract reasoning. The complexity in logical reasoning tasks lies in the correct application of logical rules. In constrast, the challenge in abstract reasoning is to identify a high-level unknown pattern from a few examples. In our work, we also provide an additional analysis using different prompting strategies and explain what kind of tasks are the most affected.
>
> We will add the discussion on these two papers and the differences with our approach to our related work.
>
> > Lack of experiment with larger models or advanced models. Fine-tuned on smaller models cannot sufficiently draw the conclusion.
> > Can you experiment with more advanced models Llama-2, with better performance than Llama1, Alpaca, or fine-tune with larger models (13B, 70B)?
>
> Yes this is a good remark, we plan to add results with the latest LLaMA2 to our paper. In our initial experiments, we used the models that were the state-of-the-art at the time and we also included in the Table 9 of appendix B.3 results for LLaMA models of larger size. We have now performed new experiments using the latest LLaMA2. Here are the results we obtained:
>
> |    | ARC$^T$  | BIG-Bench-F | Evals-S | PVR   | RAVEN$^T$-opqa |    |
> |------|------|-------|------|-------|-------|-------|
> |       |        |       |       |       | Text   | Symb  |
> | LLaMA-7B    | **0.010**| 0.012       | **0.014** | **0.060** | 0.000  | 0.000 |
> | LLaMA2-7B    | 0.005    | **0.108**   | 0.000   | 0.000 | 0.000   | **0.001** |
>
> |     | ACRE$^T$ |     | Evals-P | RAVEN$^T$-mcqa |       |
> |-------|:--------:|:-------:|:-------:|:-------------:|-------|
> |         | Text     | Symb    |      | Text    | Symb  |
> | LLaMA-7B    | 0.000    | **0.257** | **0.544**| 0.004    | 0.000 |
> | LLaMA2-7B    |**0.014** | 0.003   | 0.500   | **0.026**     | **0.149** |
> | random   | 0.33     | 0.33    | 0.5     | 0.125   | 0.125 |
>
> Fine-tuning models is a good direction but the abstract reasoning abilities of fine-tuned models are challenging to evaluate due to their memorisation ability. This is why we focused on in-context learning tasks that have not been seen during training. Per your request, we have performed additional experiments on fine-tuned LLaMA and LLaMA2 models. This remark was also raised by reviewer LbQe so we put the results in a general comment at the top. For cost reasons, we have not performed fine-tuning experiments on bigger versions of LLaMA.
>
> > The details of fine-tuning experiments, such as training data, training steps.
>
> For our fine-tuning experiments, we indicate the training set in the corresponding tables. Unless indicated otherwise, we always use a training split from the same task that is evaluated. We also evaluate the fine-tuned models on a different format (symbolic or text) than the one they have been trained on to assess if they learned abstract patterns. The training split always contains different samples than the ones used for evaluation. We fine-tune for 3 epochs on batches of size 64 with learning rate 0.0005.
>
> > Do you consider incorporating the instruction about “how to induce”, “how to deduce” into supervision?
>
> This is an interesting direction! The main challenge in our tasks is to recognise an unknown general pattern from a few examples, which is closer to abductive reasoning, so our instructions focus on the generation of this abstract pattern. We performed a few experiments in this direction in Appendix B.3 with MERIt, a model fine-tuned on logical reasoning tasks. The question of whether a model tailored for deductive or inductive reasoning can perform abstract reasoning is interesting but it is not the focus of this paper. We will consider it in our future work.

---

> > ### Author Response · Authors · 2023-11-20
> >
> > We hope that our new experiments have addressed your concerns. Please let us know if we have covered all of them, or which points you would like us to address in more detail.

---

> > > ### Comment · Reviewer_As6m · 2023-11-23
> > >
> > > Dear Authors,
> > >
> > > Thank you for providing a detailed response and additional experiments. However, as you may have acknowledged, this work, though published at a similar time as [1][2], etc., shares similar conclusions and research attempts. I do appreciate the efforts put in this direction and providing sufficient analysis. However, to be published at a high-profile conference, I do expect more insights or fundamental differences compared with prior works, which unfortunately are not clearly presented.
> > >
> > > Best,

---

### Official Review · Reviewer_LbQe · 2023-11-05

**Soundness:** 3 good
**Presentation:** 3 good
**Contribution:** 3 good
**Rating:** 6
**Confidence:** 3

**Summary:**

This paper investigates the abstract reasoning abilities of LLMs. The authors propose a benchmark for evaluating language models in order to comprehensively assess their abstract reasoning capabilities. Their experiments reveal that current LLMs struggle with abstract reasoning tasks and techniques that have previously improved performances on other NLP tasks do not result in significant enhancements for abstract reasoning.

**Strengths:**

1. The paper investigates abstract reasoning abilities of Large Language Models by creating a new benchmark combining existing datasets with novel datasets adapted from vision tasks for language models, which has not been extensively studied before.
2. The evaluation is pretty extensive including a wide range of models and tried a few techniques beyond just simple prompting.
3. The paper is well-written and organized.
4. The proposed task has not yet been solved by LLMs.

**Weaknesses:**

1. this task will be automatically solved when models of better reasoning capabilities become available.
2. The authors frame abstract reasoning as "a potential task for effective measurement of the cognitive abilities of neural models", so the utility of this benchmark is mostly evaluation of LLMs. One concern is that there isn't an actual application that would benefit from studying this kind of reasoning capabilities.

**Questions:**

1. Have authors considered fine-tuning?  It would be nice to show even fine-tuning Llama2 is not enough for solving the abstract reasoning tasks.
2. Curious to see how zephyr-7b-beta (https://huggingface.co/HuggingFaceH4/zephyr-7b-beta) performs on the proposed benchmark.
3. How is open-ended QA evaluated?
4. Do the authors have plans to maintain a leaderboard for this task? Will there be a held out test set?
5. What is the data releasing plan for this benchmark?
6. Also curious about human performance on this benchmark. For example, I couldn't figure out the example in Figure 6.

---

> ### Author Response · Authors · 2023-11-16
>
> Thank you for your constructive feedback!
>
>
>
> > This task will be automatically solved when models of better reasoning capabilities become available.
>
> Our aim in this paper is to highlight current important limitations of Large Language Models to raise awareness in the research community and explore leads to mitigate them. We believe our benchmark can be used to monitor the progress of LLMs on reasoning tasks.
>
> > The authors frame abstract reasoning as "a potential task for effective measurement of the cognitive abilities of neural models", so the utility of this benchmark is mostly evaluation of LLMs. One concern is that there isn't an actual application that would benefit from studying this kind of reasoning capabilities.
>
> We aim to measure the abilities of LLMs to reason abstractly and causally, i.e. perform reasoning on symbols invariant or robust to distribution changes. This is critical for strong generalisation and use in the real-world. Although this is not tailored to a specific application, this ability is crucial for many downstream tasks.
>
>
>
> > Have authors considered fine-tuning? It would be nice to show even fine-tuning Llama2 is not enough for solving the abstract reasoning tasks.
>
> This is an interesting direction! We aim to measure the ability of a model to build abstraction and reason on top of it but this ability is hard to measure due to the memorisation ability of LLMs. This is why we focused on in-context learning tasks that have not been seen during training. Per your request, we have performed additional experiments on fine-tuned LLaMA and LLaMA2 models. This remark was also raised by reviewer As6m so we put the results in a general comment at the top.
>
> > Curious to see how zephyr-7b-beta (https://huggingface.co/HuggingFaceH4/zephyr-7b-beta) performs on the proposed benchmark.
>
> This is an interesting question, the Zephyr model was not yet released when we conducted our initial experiments. Per your request, we tested it on our benchmark. Here are the results we obtained:
>
> |                   | ARC$^T$  | BIG-Bench-F | Evals-S | PVR   | RAVEN$^T$-opqa |       |
> |-------------------|----------|-------------|---------|-------|---------------|-------|
> |                   |          |             |         |       | Text          | Symb  |
> | Zephyr-7B-$\beta$ |   0.015  |   0.292     | 0.043   | 0.209 |   0.009  |   0.145    |
>
>
> |                     | ACRE$^T$ |         | Evals-P | RAVEN$^T$-mcqa |       |
> |---------------------|:--------:|:-------:|:-------:|:-------------:|-------|
> |                     | Text     | Symb    |         | Text          | Symb  |
> | Zephyr-7B-$\beta$   |   0.106  |  0.516  | 0.504   | 0.000         | 0.022 |
> | random              | 0.33     | 0.33    | 0.5     | 0.125         | 0.125 |
>
>
> > How is open-ended QA evaluated?
>
> For the open-ended QA tasks, we ask the LLMs to generate the expected output pattern. We use regular expressions to catch the pattern in the model's answer as it may not follow our instructions and wrap its answer in some text (e.g. reasoning justifications or "The answer is ..."). We consider the answer correct even if the format of the answer diverges from the instructions. Before the evaluation, we performed multiple trials with each model to capture all the possible formats they used and we included them into our regular expressions. We did sanity checks to ensure we did not miss any formats they could use.
>
> > Do the authors have plans to maintain a leaderboard for this task? Will there be a held out test set?
> > What is the data releasing plan for this benchmark?
>
> This is a good suggestion! We did not plan to create a leaderboard but we will look into the idea and see if its is feasible. We currently have no heldout test set but we will consider creating one. We currently plan to release our code and data in open-source along with the paper to ease reproducibility and evaluation with new models.
>
> > Also curious about human performance on this benchmark. For example, I couldn't figure out the example in Figure 6.
>
> Sorry for the confusion, Figures 5 and 6 are the same example in text and symbolic formats. We will clarify this point in the paper. This example illustrates a backward-blocking case. From the input examples, we aim to determine if the red sphere causes the light activation. The red sphere is never observed alone in the examples, it is only seen with the yellow cube, so we cannot determine which one causes the light activation. So from the data, we cannot determine if the light should be on or off in the test case.

---

> > ### Author Response · Authors · 2023-11-20
> >
> > We hope that our new experiments have addressed your concerns. Please let us know if we have covered all of them, or which points you would like us to address in more detail.

---

### Official Review · Reviewer_2kzP · 2023-11-19

**Soundness:** 2 fair
**Presentation:** 3 good
**Contribution:** 2 fair
**Rating:** 5
**Confidence:** 4

**Summary:**

This paper proposes to evaluate the abstract reasoning ability of LLMs by curating a set of datasets. Overall the authors show that the performance of current LLMs are limited and various techniques do not help.

**Strengths:**

- The paper is well written and easy to follow
- The curated benchmark seems high quality
- The experiments are extensive and demonstrate the main point.
- The observation that basic techniques do not improve performance is significant.

**Weaknesses:**

- This new benchmark introduced are largely existing datasets thus with limited novelties. There are also existing works on evaluating the inductive reasoning ability of LLMs such as https://arxiv.org/pdf/2306.09841.pdf.
- This paper does not evaluate slightly more complicated prompting methods, such as simply generating more samples of code and filter by number of training examples passed. Existing papers proposing more complicated pipelines: https://arxiv.org/pdf/2212.10923.pdf, https://arxiv.org/abs/2309.05660 ,https://arxiv.org/abs/2310.08559

**Questions:**

n/a

---

> ### Author Response · Authors · 2023-11-20
>
> Thank you for your constructive feedback!
>
> > This new benchmark introduced are largely existing datasets thus with limited novelties. There are also existing works on evaluating the inductive reasoning ability of LLMs such as https://arxiv.org/pdf/2306.09841.pdf. ([1])
>
> Our study has been independently conducted and released at the same time as [1]. The authors measure the logical reasoning abilities of LLMs on standard benchmarks. While this is related to our work, the paper and the datasets used measure an ability different from abstract reasoning. The complexity in logical reasoning tasks lies in the correct application and extraction of logical rules. In constrast, the challenge in abstract reasoning is to identify a high-level unknown pattern from a few examples. In our work, we also provide an additional analysis using different prompting strategies and explain what kind of tasks are the most affected based on multiple factors (e.g. input dimensionality, causal reasoning needed). We also include in our experiments GPT-4, LLaMA, and LLaMA2 in the latest version. We will add the discussion on this paper and the differences with our approach to our related work.
>
> [1] Xu, Fangzhi, et al. "Are Large Language Models Really Good Logical Reasoners? A Comprehensive Evaluation From Deductive, Inductive and Abductive Views." arXiv preprint arXiv:2306.09841 (2023).
>
>
> > This paper does not evaluate slightly more complicated prompting methods, such as simply generating more samples of code and filter by number of training examples passed.
>
> In our early experiments, we found that LLMs struggle to extract the correct abstract pattern and refinement techniques that ask the model to correct its answer did not solve this initial issue. Therefore, we focused our work on prompting techniques that have been shown to significantly improve the performance of LLMs, from their first response onwards, such as chain-of-thought and program-of-thought. Per your request, we will perform new experiments with code refinement. Given the short amount of time, we will do our best to post our results in a comment before the end of the discussion period. In all cases, we will integrate the results to the paper.
>
> Please, let us know if this answers your concerns.

---

> > ### Author Response · Authors · 2023-11-26
> >
> > Thank you for your patience! As requested, we have conducted new experiments using four more complex prompting techniques. Code-filtering consists of generating multiple code responses and filtering out the programs that cannot solve the example cases. Code-refinement (similarly to [1,2]) is an iterative process in which the model generates a first program. The programme is run on the examples and, if not all answers are correct, the model is prompted to correct its answer based on the output of the interpreter. Self-filtering and self-refinement (similarly to [2,3]) are similar techniques, but ask the LLM to assess whether the given answer is correct, rather than relying on an interpreter. We conducted experiments on BIG-Bench-F and PVR using GPT-3.5 and GPT-4. We used the latest versions of GPT-3.5-Turbo (gpt-3.5-turbo-0613) and GPT-4-Turbo (gpt-4-1106-preview).
> >
> > [1] Wang, Ruocheng, et al. "Hypothesis search: Inductive reasoning with language models." arXiv preprint arXiv:2309.05660 (2023).
> >
> > [2] Qiu, Linlu, et al. "Phenomenal Yet Puzzling: Testing Inductive Reasoning Capabilities of Language Models with Hypothesis Refinement." arXiv preprint arXiv:2310.08559 (2023).
> >
> > [3] Madaan, Aman, et al. "Self-refine: Iterative refinement with self-feedback." arXiv preprint arXiv:2303.17651 (2023).
> >
> > |                        | BIG-Bench-F | PVR   |
> > |------------------------|-------------|-------|
> > | GPT-4-Turbo-code              | 0.280       | **0.152** |
> > | GPT-4-Turbo-code-filtering       | **0.400**       | **0.152** |
> > | GPT-4-Turbo-code-refinement      | 0.296       | 0.144 |
> > | GPT-4-Turbo                     | 0.268       | 0.000 |
> > | GPT-4-Turbo-self-filtering      | 0.284       | 0.004 |
> > | GPT-4-Turbo-self-refinement     | 0.252       | 0.000 |
> > | GPT-3.5-Turbo-code            | 0.316       | **0.200** |
> > | GPT-3.5-Turbo-code-filtering     | 0.352       | **0.200** |
> > | GPT-3.5-Turbo-code-refinement    | 0.336       | 0.188 |
> > | GPT-3.5-Turbo                   | 0.416       | 0.116 |
> > | GPT-3.5-Turbo-self-filtering    | **0.444**       | 0.124 |
> > | GPT-3.5-Turbo-self-refinement   | 0.323       | 0.084 |
> >
> >
> > We also ran additional experiments with GPT-3.5-Turbo. We varied the number of generations for the filtering methods and the maximum number of refinement steps for the refinement methods.
> >
> > |                              | BIG-Bench-F | BIG-Bench-F | BIG-Bench-F |           | PVR   | PVR   | PVR   |
> > |------------------------------|-------------|-------------|-------------|-----------|-------|-------|-------|
> > |                              | 2 steps     | 4 steps     | 8 steps     |           | 2 steps | 4 steps | 8 steps |
> > | GPT-3.5-Turbo-code-filtering   | 0.320       | 0.352       | **0.380**   |           | 0.200 | 0.200 | **0.208** |
> > | GPT-3.5-Turbo-code-refinement | 0.320       | **0.336**   | 0.335       |           | 0.188 | 0.188 | **0.201** |
> > | GPT-3.5-Turbo-self-filtering   | 0.428       | **0.444**   | 0.424       |           | 0.132 | 0.124 | **0.148** |
> > | GPT-3.5-Turbo-self-refinement  | **0.364**   | 0.323       | 0.307       |           | **0.112** | 0.084 | 0.080 |
> >
> > The performance obtained with GPT-4-Turbo is lower than the one obtained with GPT-4 in our original experiments, while the performance of the latest GPT-3.5-Turbo-0613 is higher. However, the accuracy does not exceed the one observed in our initial experiments. We can also observe that filtering from independent queries yields greater improvements than refining a single query multiple times. Code filtering generally performs better than self-filtering, except for GPT-3.5-Turbo on BIG-Bench-F. Finally, increasing the number of generations and refinement steps generally improves performance, except for self-refinement, where increasing the number of steps decreases performance. The latter is probably due to an accumulation of errors during the different generation processes.
> >
> > We hope that our new experiments have addressed your concerns. Please let us know if we have covered all of them, or which points you would like us to address in more detail.

---

### Author Response · Authors · 2023-11-16
**Additional Fine-tuning Experiments**

The abstract reasoning abilities of models fined-tuned on a specific task are challenging to evaluate as fine-tuned models might exploit spurious biases that are specific to the task instead of abstract mechanisms. This issue has been raised by [1].
For this reason, we initially chose to focus on in-context learning tasks that have not been seen during training. This is also the privilegied approach in recent related works [1,2].

Per your requests, we have conducted new fine-tuning experiments. For this series of experiments, we use splits with task and distribution variations. At this time, we have conducted fine-tuning on LLaMA and LLaMA2 on the training sets of the ARC$^T$, ACRE$^T$, and RAVEN$^T$ datasets. We are also conducting experiments on the PVR dataset and they will be integrated when they are completed. Fine-tuning does improve the model performance on the initial set for ACRE and RAVEN but this does not necessarily correlate with an increased abstract reasoning capacity as the performance does not systematically transfer to the split variations, in particular for RAVEN. We will include these results and a deeper analysis in the paper. We can also note that fine-tuning does not significantly improve the model performance for ARC.

Accuracy of fine-tuned models on the ARC dataset:
|                 | ARC$^T$      |
|-------------|--------------|
| LLaMA-7B-AR-LoRA*    | 0.018        |
| LLaMA2-7B-AR-LoRA*   | 0.010        |

Accuracy of fine-tuned LLaMA-7B on the ACRE dataset using LoRA. Rows represent the dataset on which the model is fine-tuned, and columns represent the dataset on which the model is evaluated. We perform mexperiments on two additional splits: the compositional set contains different combinations of inpus than the ones seen in the training set and the systematic set contains a different distribution. The best result for each dataset in indicated in **bold**.
| LLaMA-7B-AR-LoRA$^*$ |         | ACRE$^T$-Eval |        | -Comp |        | -Sys |   |
|--------------|--------|--------|---------------|--------|--------|-------|--------|
|                       |        | Text          | Symb   | Text   | Symb  | Text   | Symb   |
| ACRE$^T$-Train        | Text   | **0.755** | 0.614       | **0.741** | 0.606 | **0.727** | 0.550 |
|                       | Symb   | 0.081  | **1.000**     | 0.102  | **0.999** | 0.095 | **0.999** |

Accuracy of fine-tuned LLaMA2-7B on the ACRE dataset using LoRA:
| LLaMA2-7B-AR-LoRA$^*$ |        | ACRE$^T$-Eval |        | -Comp |         | -Sys |   |
|--------------|--------|--------|------------|--------|--------|-------|--------|
|                       |        | Text          | Symb   | Text   | Symb  | Text   | Symb   |
| ACRE$^T$-Train        | Text   | **0.997** | 0.662       | **1.000** | 0.651 | **0.994** | 0.626 |
|                       | Symb   | 0.568  | **1.000**     | 0.579  | **1.000** | 0.539 | **0.999** |

Accuracy of fine-tuned LLaMA-7B on the RAVEN dataset using LoRA. We perform experiments on two additional splits: the four set contains longer patterns with four figures instead of one and the in-center set contains a different organisation of input figures:
| LLaMA-7B-AR-LoRA$^*$ |         | RAVEN$^T$-mcqa-Eval |        | -Four |        | -In-Center |        |
|-------------|--------|--------|----------|--------|--------|-------|--------|
|                       |        | Text               | Symb   | Text   | Symb  | Text   | Symb   |
| RAVEN$^T$-mcqa-Train  | Text   | **0.558** | 0.322              | **0.050** | 0.168 | 0.000 | 0.010 |
|                       | Symb   | 0.232  | **0.460**          | 0.014  | **0.287** | **0.002** | **0.016** |

Accuracy of fine-tuned LLaMA2-7B on the RAVEN dataset using LoRA:
| LLaMA2-7B-AR-LoRA$^*$ |        | RAVEN$^T$-mcqa-Eval |        | -Four |        | -In-Center |     |
|--------|--------|--------|-------------|--------|--------|-------|--------|
|                       |         | Text               | Symb   | Text   | Symb  | Text   | Symb   |
| RAVEN$^T$-mcqa-Train  | Text   | **0.977** | 0.694              | **0.557** | **0.522** | 0.536 | **0.085** |
|                       | Symb   | 0.965  | **0.938**          | 0.498  | 0.442 | **0.767** | 0.064 |

[1] Jin, Zhijing, et al. "Can Large Language Models Infer Causation from Correlation?." arXiv preprint arXiv:2306.05836 (2023).

[2] Wu, Zhaofeng, et al. "Reasoning or reciting? exploring the capabilities and limitations of language models through counterfactual tasks." arXiv preprint arXiv:2307.02477 (2023).

---

### Meta-Review · Area_Chair_68mD · 2023-12-12

**Metareview:**

The paper is generally well-written and organized, and the authors make efforts to evaluate abstract reasoning in Large Language Models (LLMs) with an extensive set of benchmarks and experiments. Despite the authors' efforts to address the concerns raised and conduct additional experiments, the reviewers maintain their stance that the paper falls marginally below the acceptance threshold, citing limitations in novel contributions and practical applications. The additional experiments, while valuable, do not sufficiently elevate the paper's status within the context of ample existing literature on the abstract reasoning abilities of LLMs.

Considering all reviewers' critiques, the reiterated concerns about novelty, and the authors' responses, it appears that the paper does not meet the high bar for originality and impact expected at this conference. Although the submission is well-written and constitutes a thorough investigation, it lacks the innovative edge and does not sufficiently diverge from work already available in the field. The authors are encouraged to refine their benchmarking approach further, seek fundamental insights, and articulate a clearer differentiation from existing literature in future submissions.

**Justification For Why Not Higher Score:**

Despite the authors' efforts to address the concerns raised and conduct additional experiments, the reviewers maintain their stance that the paper falls marginally below the acceptance threshold, citing limitations in novel contributions and practical applications. The additional experiments, while valuable, do not sufficiently elevate the paper's status within the context of ample existing literature on the abstract reasoning abilities of LLMs.

**Justification For Why Not Lower Score:**

N/A

---

### Decision · Program_Chairs · 2024-01-16

Reject